# Infant-Feeding Patterns and the Effects of Early Introduction to Formula and Solid Foods on Childhood Overweight or Obesity among 10-Years-Olds in a Low Socioeconomic Area of Lebanon: An Exploratory Analysis

**DOI:** 10.3390/children9071030

**Published:** 2022-07-11

**Authors:** Haider Mannan, Yonna Sacre, Randa Attieh, Dany Farah

**Affiliations:** 1Translational Health Research Institute, Western Sydney University, Campbelltown, NSW 2560, Australia; 2Department of Nutrition and Food Sciences, Holy Spirit University of Kaslik, Jounieh P.O. Box 446, Lebanon; yonnasacre@usek.edu.lb (Y.S.); randaattieh@usek.edu.lb (R.A.); dany.j.farah@net.usek.edu.lb (D.F.)

**Keywords:** obesity, childhood, risk factors, timing, formula, solids

## Abstract

In Lebanon, there has been an alarming increase in childhood overweight and obesity. In addition, most mothers do not meet the WHO recommendation that infants should be introduced to formula or solids only during the second half of their first year. Because the study population, Burj Hammoud, which is a low socioeconomic district, is small, we performed an exploratory analysis of infant feeding patterns and the effects of introducing formula and solids within six months, respectively, on childhood overweight/obesity among 10-year-old children. A total of 101 mothers were recruited from seven intercommunity dispensaries located across the district. Descriptive, univariate and multivariate logistic regression analyses were performed. There were 86.1% infants initiating breastfeeding, 18% exclusively breastfeeding at 6 months of age, 67.1% and 52.6% starting formula and solids by 6 months, respectively, and 53.5% becoming overweight/obese by the age of 10 years. Working mothers were significantly less likely to introduce formula but not solids within the first six months of infancy. Based on two case-control studies, after controlling for maternal employment, there were 2.278- and 1.511-fold significantly higher odds of introducing solids and formula before the age of 6 months compared with after the age of 6 months, respectively, for the overweight/obese individuals among 10-year-olds. Future research should focus on conducting a larger study by incorporating other low socioeconomic regions to confirm these relationships.

## 1. Introduction

Infant feeding practices are modifiable postnatal factors that influence body weight in later life [1]. In many countries, mostly those of high- and middle-income levels, it was found that infants introduced to formula or solid foods within the first six months have a higher likelihood of being overweight or obese during childhood [2,3,4]. The WHO recommends breastfeeding within the first hour of life, exclusive breastfeeding for the first six months and, thereafter, infants receive complementary foods with continued breastfeeding up to two years or beyond [5]. In Lebanon, breastfeeding exclusivity below the age of six months [6] was only 15%, despite a high rate (95.4%) of breastfeeding initiation [7]. Hence, most children do not meet the WHO recommendations for exclusive breastfeeding in the first six months of their infancy. Mothers of upper socioeconomic status (SES) were significantly less likely to exclusively breastfeed for six months [8]. Furthermore, childhood overweight and obesity has increased alarmingly in recent years, reaching 32.1% in 2019, of which 10.9% were obese [9].

In this exploratory study, we investigate the patterns of infant feeding, the relationship between maternal SES and the early introduction to formula and solid feeding, respectively, and the effects of the latter two variables on overweight or obesity among 10-year-olds in Burj Hammoud, which is a low socioeconomic district in Lebanon. By “infant feeding patterns” we mean the dietary patterns of infants, which include breastfeeding, formula, complementary feeding, such as semi-solid and solid foods. The timing of introduction to these specific diets is usually not of concern when we discuss infant feeding patterns. In contrast, by the “early introduction to formula and solid foods” we mean infants who were introduced to formula and solid foods before their WHO recommended age of six months. To our knowledge, this is the first study in Lebanon to examine whether early introduction to formula and solid feeding, respectively, are risk factors of childhood overweight or obesity. We hypothesize that both are associated with their SES. As the primary analytical research question, we hypothesize that children who are introduced to infant formula and solid foods in the first six months are more likely to be overweight or obese when they are 10 years old. We also assess the data collection instrument and recruitment strategy for this study with the intention of helping improve them for similar studies in other low socioeconomic settings.

## 2. Materials and Methods

We conducted an exploratory study between July 2019 and September 2019 in Burj Hammoud, which is considered one of the poorest and most densely populated districts in the Middle East [10]. It is located in Mount Lebanon, which is adjacent to Beirut. We used convenience sampling to select the mothers for this study. The participating mothers were clients of seven intercommunity dispensaries located across Burj Hammoud. These dispensaries were contacted for their consent to help recruit mothers for the survey and all agreed to do so. In Lebanon, infant formula for those younger than one year is sold only at dispensaries, as such formula is categorized as a pharmaceutical product; for this reason, people with infants under one-year old must purchase formula from dispensaries. The selection criteria for our survey were that mothers must be of Lebanese descent and currently have a 10-year-old child who, at birth, was a singleton, full term, normal weight and healthy. A multi-component questionnaire was administered to the participating mothers.

We did not require a large sample size to investigate our primary analytical research question. Sample size estimations during the planning stage of the study showed that at least 100 participants would be adequate to detect significant effects for both introduction to formula and solid foods within the first 6 months, respectively, on the odds of childhood overweight/obesity, after adjusting for several potential confounders and fixing power and type 1 errors at 80% and 5%, respectively. To account for likely non-responses, we aimed to recruit 120 women; the final number of women who met the selection criteria and agreed to participate totaled 101, thus fulfilling the minimum sample size requirement.

Data for infant feeding histories were collected retrospectively by asking mothers to recall information 10 years after their child’s birth. Those who hesitated to provide a response and did not confirm their response were excluded. The weight and length of children were not measured by anthropometric methods but by using reports given by mothers. Issues regarding recall pertaining to these measures are discussed in Section 4. We calculated each ten-year-old’s BMI from his/her weight and height. If a child’s BMI was at least equal to the 85th percentile of the distribution of BMI, he/she was classified as overweight or obese.

Maternal educational level (University complete = 1 vs. No University = 0) and return to work were considered because they correlated with a mother’s SES. Crowding index per room was considered because it is widely used as a metric for crowding, and it correlates negatively with SES [11]. The World Health Organization (WHO) considers crowding a major factor affecting human health as the crowding of persons within a small space promotes the rapid spread of infection, especially for children [12]. Overcrowding also causes many social problems, such as lack of privacy and possible sexual harassment and can, therefore, act as a barrier to breastfeeding. The crowding index per room was calculated by dividing the number of individuals, including the participant but excluding newborns, living in a room per household, by the number of rooms in the house except for kitchen and bathrooms. A value greater than 1 is an indicator of crowding and greater than 1.5 an indicator of severe crowding [13]. The distribution of the crowding index was non-normal, based on the Shapiro–Wilk test; we, therefore, used the median to find its measure of central tendency.

Both types of analysis were performed—descriptive and analytical (univariate and multivariate logistic regressions). Univariate logistic regression analysis was performed to determine the individual effect of the crowding index, maternal education and maternal return to work following maternity leave, on age at introduction to solid foods by 6 months versus over 6 months, and age at introduction to formula feeding by 6 months versus over 6 months. Using two separate case-control studies, multivariate logistic regression analyses were performed to determine the effect of overweight or obesity among 10-year-olds on the odds of introducing formula and solids within the first 6 months, respectively, after adjusting for maternal return to work; this latter was found to be an operational confounder when examining the relationship. Other potential confounders, such as crowding index and maternal education, were not found to be operational confounders. Maternal age was not found to be an operational confounder and was not included in the model. The operational definition of confounding requires a stronger risk adjustment compared with that of the classical definition [14]. There were insufficient data to calculate children’s age at breastfeeding cessation and body mass index during pregnancy so we could not assess whether these variables were operational confounders. For sensitivity analysis, we adjusted for all four covariates (maternal return to work, maternal education, crowding index and maternal age) in our logistic model and found that the result did not change from that which we reported in this article that adjusted only for maternal return to work (results not shown). There was no collinearity between introducing solid feeding within the first 6 months and introducing formula feeding within the first 6 months as their variance inflation factors were only 1.57 and 1.59, respectively, well below the cut point of greater than 10. The statistical tests and 95% confidence intervals for the odds ratios, were calculated using the Wald method. Data were analyzed using SAS version 9.4 [15].

## 3. Results

### 3.1. Crowding Index, Education, Return to Work

The median crowding index was 1.33, indicating that 50% of dwellings had more than 1.33 persons living per room. In fact, 59.4% of the dwellings had more than 1 person living per room and, hence, were crowded, while 35% of the dwellings had more than 1.5 persons living per room and, hence, were severely crowded. Only 30% of mothers had completed a university education, while the remaining 70% either had no education (6%) or only elementary (44%) or secondary (20%) education. A total of 64.1% mothers returned to work after their maternity leave. These results are not presented here.

### 3.2. Reasons for Returning to Work

According to Figure 1, the main reason for mothers returning to work after their maternity leave was ‘financial’ (14.9%), followed by ‘maintaining a career’ (10%). Only 3% stated ‘[they] needed [an] outlet outside home’ and only 1% stated ‘job-related benefits’.

### 3.3. Support for Breastfeeding at Work

Out of a total of 29 mothers who returned to work, most (55.2%) described their workplace as not supportive at all for breastfeeding; of the mothers who returned to work, the most common response (33.7%) was not receiving any information regarding breastfeeding during their antenatal visits (Figure 2).

### 3.4. Child Overweight/Obesity and Infant Feeding Patterns

The following results are not shown in a table or figure: 53.5% of the children became overweight or obese by 10 years of age. Breastfeeding initiation was undertaken by 86.1% of mothers, and exclusive breastfeeding for the first six months was undertaken by 18%. By the ages of 4 and 6 months, 19.9% and 52.6% infants received solid foods, respectively, while 48.6% and 67.1% received formula, respectively. In addition, 39.1% of mothers weaned their infants within the first 6 months, and 65.5% before 1 year.

### 3.5. Reasons for Breastfeeding Cessation

The main reason for stopping breastfeeding was insufficiency of breastmilk (28.7%), followed by baby weaning himself/herself (24.8%), fatigue (10.9%), sore nipples/engorged breast (4%), return to work (4%), difficulty with breastfeeding techniques (3%), maternal illness (3%), physicians telling [them] to stop breastfeeding (3%), formula feeding preferable (3%) and post-natal depression (2%); the remaining 15.6% did not respond.

### 3.6. Feeding Preferences for Children by Mothers and Antenatal Visits

Results showed that almost three-fourths (78.2%) of all mothers agreed that breastfeeding only is the best way to feed the infant during the first six months of life, followed by both breast and formula feeding (19.8%). None stated formula feeding only as the best way to feed the infant during the first six months of life.

### 3.7. Timings of Initiating Formula or Solids and Breastfeeding Cessation

There were 41.6% mothers who agreed with introducing formula or solid foods by the age of 6 months to 1 year, 18.8% agreed with introducing formula or solids above the age of 1 year, 17.8% by the age of 5 to 6 months and 16.8% by the age of 3 to 4 months. Most (55.4%) wanted to completely stop breastfeeding by the age of 12 months, 37.6% by the age of 6 to 12 months, and only 5% and 2% between the ages of 5 and 6 months and 3 and 4 months, respectively.

### 3.8. Effects of Return to Work, Education and Crowding Index on Odds of Initiating Formula and Solids Feedings within Six Months

Table 1 below shows that mothers who returned to work after their maternity leave have a 69% and significantly lower odds of introducing formula feeding within the first six months compared with those who did not return to work. Mothers with a secondary and higher education have a 58.9% but non-significantly higher odds of introducing formula feeding within the first six months compared with those who have no education or primary education. For every unit increase in crowding index, there is a 22.8% higher odds of starting formula feedings within the first six months, but the effect is not significant. Mothers who returned to work after their maternity leave have a 2.95-fold and significantly higher odds of starting solid feeding within the first six months compared with mothers who do not return to work. Mothers with a secondary or higher education have a 27.2% but non-significantly higher odds of introducing solid foods within the first six months compared with those who have no education or primary education only. For every unit increase in crowding index, there is a 17.5% higher odds of starting solid foods within the first six months, but the effect is non-significant.

### 3.9. Effects of Initiating Formula and Solid Feedings within Six Months on Odds of Overweight/Obesity

Logistic regression, after adjustment for confounding by maternal return to work following maternity leave (see Table 2), showed that there are 2.278 and 1.511-fold higher odds of introduction to formula and solid foods by the of age six months compared with above the age of six months associated with overweight or obesity among 10-year-olds, respectively. Both the effects are significant and, using Rosenthal’s criteria [16], can be classified as moderate. There is a positive relationship between maternal return to work following maternity leave and the odds of introduction to formula and solid foods by the age of six months as compared with above the age of six months, respectively.

## 4. Discussion

In this study, the participants came from a low socioeconomic district in Lebanon—Burj Hammoud—and were mostly of the same background as determined by their low education level and high crowding index per room. The sample was, thus, representative of the study population in this regard. There were 52.6% and 67.1% infants receiving solid foods and formula by the age of six months, respectively. A total of 39 mothers and 65.5 percent of mothers stopped breastfeeding by the time their child rached the age of six months and before one year, while 19.9% introduced solid foods by the age of four months. A study conducted among Lebanese adolescents comprising both low and middle socioeconomic groups showed these percentages to be higher, e.g., 41%, 70% and 41.6%, respectively [17]. The most common main reason for stopping breastfeeding in our sample was insufficiency of breastmilk (28.7%), closely followed by baby weaning himself/herself (24.8%). In addition, most or 55.2% of working mothers did not receive any workplace support at all for breastfeeding, out of which the most common (33.7%) were mothers who did not receive information regarding breastfeeding practices during the antenatal visits, thus supporting the view that in Lebanon there are no structured activities, or interventions embedded within the health care system that specifically provide information and support for breastfeeding mothers [18]. Briefly, the implications of our findings are that most mothers in Burj Hammoud do not follow the WHO guidelines of not starting formula and solid feeding within the first six months of infancy. Breastfeeding exclusivity among working mothers can be increased by enhancing workplace support for breastfeeding and increasing its awareness among those not receiving any workplace support for breastfeeding.

Mothers with more education had a higher odds of introducing infant formula and solid foods by the age of six months. Although the effects were not significant, their directions agree with the study by Batal et al. (2010) who found a significant effect [19]. Formula feeding is highly promoted by the media in Lebanon and during the time of this study it was subsidized. However, later when there was an economic crisis, subsidies were lifted for children between one and three years old [20].

Maternal return to work following maternity leave had significantly lower odds of introducing formula feeding within the first six months. This agrees with the study by Hamade et al. (2013), except that the timing examined was within three months [21]. Furthermore, we found that mothers returning to work following maternity leave had almost a threefold higher odds of introducing solid foods to their infants within the first six months. This agrees with a large-scale cross-sectional study across Lebanon, which found that women employed outside the home were almost twice as likely to introduce solid foods before the age of four months [22]. Mothers living in more crowded households had higher odds of introducing formula feeding but lower odds of introducing solid foods within the first six months, but they were not significant.

Our results agree with the literature which shows that infants having an early introduction to formula or solid foods have a higher likelihood of being overweight or obese by the time they are 10 years old [7,8]. There was also a positive non-significant relationship between maternal return to work following maternity leave and the odds of being overweight or obese among 10-years-olds. This finding agrees with several previous studies, apart from the fact that the effect was significant and the studies were not for 10-year-olds [23,24]. Several theoretical perspectives, such as working mothers spending less time in meal preparation and relying on fast and readily prepared food, and children spending less time participating in physical exercise due to lack of parental time available to drive them to activities, underlie potential linkages between maternal employment and childhood overweight or obesity [23].

An important strength of this study is that the primary research question, which was to examine the effect of the early introduction to infant formula and solid foods, on childhood overweight and obesity among 10-years-olds, has never previously been studied in Lebanon, or in any low socioeconomic area of Lebanon. In addition, data were collected using a simple and easy to understand questionnaire completed by the interviewer, which limits bias if the participant misunderstood or transcribed information in different ways.

The convenience sample helped us gather useful data and information that would have been more difficult using probabilistic sampling techniques, as they require more formal access to lists of sampling units. Importantly, we found that most participants had a lower educational background and belonged to crowded households, so were likely to belong to lower socioeconomic class and, in that regard, appropriately represented the Burj Hammoud population.

To better evaluate the validity of our study, we considered both the statistical and clinical significance of our findings [24] as *p*-values are highly dependent on sample size [25]. Hence, we also calculated effect size, which is independent of sample size and useful for comparing results from different studies [26]. We did not examine effect modifying groups for the relationship between the early introduction to formula and solid food and the odds of overweight or obesity among 10-year-olds, respectively, as identifying them was not an objective during the study planning stage.

A previous study showed that maternal recall of breastfeeding history of their children can be very accurate even 20 years after their birth [27]. Since the maternal recall period for our study was only 10 years, any underestimation of our observed associations due to maternal recall may be small. To minimize this further, we excluded eligible mothers who did not confirm their response with regard to their children’s breastfeeding history after being asked to do so. There was no recall involved for mothers providing a response to the question about weight and length of their children because these children were already 10 years old when their mothers were asked these questions.

The measure of crowding index per room which we used to measure crowding, although widely used, has several limitations. It assumes that children and adults have similar space requirements, which may not be accurate for all cultures. Furthermore, as this measure simply compares the number of people and number of rooms, it does not consider additional factors that may affect crowding, e.g., the age and sex of household members and the need for individual space.

The inclusion criteria were highly specific, which made it time consuming to recruit participants from the dispensaries covering different areas of Burj Hammoud. Some dispensaries mentioned that it was hard to recruit mothers of Lebanese descent at their centers, while others worked more with elderly people. The recruitment of mothers was conducted by a member of staff standing in the waiting room of doctors’ clinics in the dispensaries, which sometimes made communication difficult because it was a noisy environment. To overcome these logistical issues, for data collection in future surveys, it would be more convenient to engage with schools where mothers of 10-year-olds can be more easily found and to measure the children’s weight and height using anthropometric methods. Alternatively, Google form surveys could be considered as this would enable the collection of data easily and quickly, with the ability to export into Excel. This would help mothers to respond to questions on infant feeding practices in the privacy of their homes. Data should be collected from mothers living in different low socioeconomic areas of Lebanon to confirm the findings of this exploratory study.

## 5. Conclusions

Early supplementations of formula and solid feeding were found to be risk factors for childhood overweight or obesity among 10-years-olds in Burj Hammoud. Future research should focus on conducting a larger study involving several low socioeconomic regions of Lebanon to investigate these relationships further.

## Figures and Tables

**Figure 1 children-09-01030-f001:**
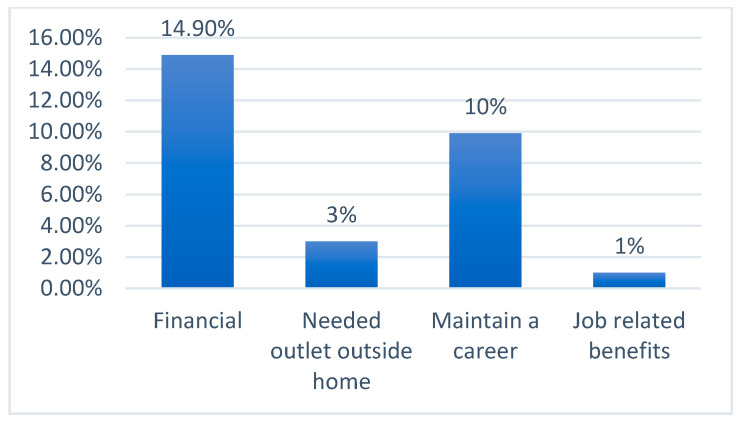
Reasons for going back to work.

**Figure 2 children-09-01030-f002:**
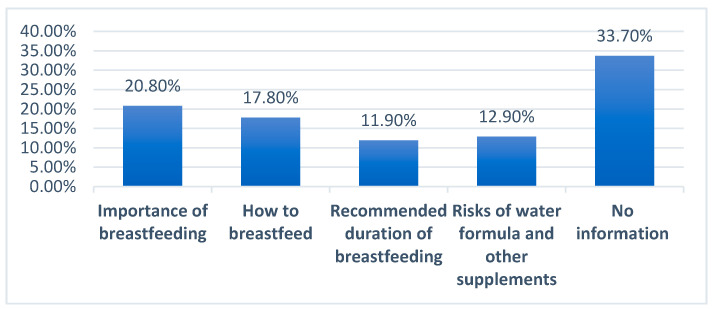
Information given during the antenatal visits.

**Table 1 children-09-01030-t001:** Effect of maternal education, maternal employment and crowding index on age at introduction to formula feeding by 6 months versus over 6 months.

	**Formula Feeding Introduced at Age ≤ 6 Months** **Odds Ratio, *p* Value** **(95% CI)**
Maternal education	-
(No education/elementary)	0.364, 1.589
Secondary and higher	(0.584, 4.321)
Crowding Index	0.559, 1.228 (0.616, 2.448)
Maternal return to work	-
(No)	0.000, 0.310 *
Yes	(0.372, 0.971)
	**Solid Foods Introduced at Age ≤ 6 Months** **Odds Ratio, *p* Value** **(95% CI)**
Maternal education	-
(No education/elementary)	0.628, 1.272
Secondary and higher	(0.481, 3.365)
Crowding Index	0.524, 1.175 (0.716, 1.931)
Maternal return to work	-
(No)	0.049, 2.945 *
Yes	(1.004, 8.639)

Note: The statistical tests and 95% confidence intervals are based on the Wald method for logistic regression; * indicates *p* ≤ 0.05.

**Table 2 children-09-01030-t002:** Estimated odds ratios, 95% confidence intervals (CI) and *p*-value based on Wald Z statistic for the effect of overweight/obesity among 10-year-olds on odds of age at introduction to formula and age at introduction to solid foods, respectively, after controlling for maternal return to work following maternity leave, in Burj Hammoud, Lebanon.

Dependent Variable	Odds Ratio, *p* Value	95% CI
Age at starting solid foods		
≤6 months vs. >6 months	2.278 *, 0.015	1.170–4.437
Age at starting formula		
≤6 months vs. >6 months	1.511 *, 0.030	1.040–2.195

Notes: The statistical tests and 95% confidence intervals are both based on Wald method. The model controlled for the effect of maternal return to work; the superscript * indicates *p* ≤ 0.05.

## Data Availability

Not applicable.

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
