# Peer review of "Infant-Feeding Patterns and the Effects of Early Introduction to Formula and Solid Foods on Childhood Overweight or Obesity among 10-Years-Olds in a Low Socioeconomic Area of Lebanon: An Exploratory Analysis"

_children, 2022, doi:10.3390/children9071030_

Round 1

Reviewer 1 Report

In their cross-sectional study, the Authors performed an exploratory analysis of infant feeding patterns and effects of introducing formula and solids within six months on childhood overweight/obesity among 10-year-old. Of interest, they found that 86.1% infants initiating breastfeeding, 18% exclusively breastfeeding 6 months, while 67.1% and 19 52.6% starting formula and solids by 6 months, and 53.5% becoming overweight/obese by 10-year age. Working mothers were significantly less likely to introduce formula but not solids within 6 months, and after controlling for mother’s employment, there were 4.40- and 4.37-folds significantly higher odds of overweight/obesity for 10-year-old when they initiated solids and formula within 6 months compared to over 6 months, respectively.

The study is interesting and with a good research question. Methodology is acceptable, references updated, and readability good. Fine English revision should be performed.  Strengths and limitation (i.e., small sample size, recall bias…) not only have been listed by the Authors, but also discussed.

As suggestion, to remove decimals from Fig. 2.  

Author Response

Reviewer reports:

I found this paper presents important information that adds to the infant-feeding patterns knowledge pool, and its effect on childhood weight status among 10-years-olds in low socioeconomic status region.
I have few comments to be considered on the revised version

  1. Principal comments:

Title:

  • Please explain the difference between “infant feeding patterns” and “early introduction to formula and solid foods”. The meaning of infant-feeding patterns is not defined on this revised version.

Response: By “Infant feeding patterns” we mean the dietary patterns of infants which include breastfeeding, formula, complementary feeding like semi-solid and solid foods. The timing of introduction to these specific diets is usually not of concern when we discuss infant feeding patterns. In contrast, by “early introduction to formula and solid foods” we mean infants who were introduced to formula and solid foods before their WHO recommended age of six months. The reference for the WHO recommendation in regards to this was already provided in our paper. The above changes are now added in the text which are highlighted in yellow.

Materials and method:

  • In the material and method section, it is suggested to divide into paragraphs according to the “study population,” “the process of collecting information,” “covariate,” “statistical analysis” and so on.

Response: We have now divided the material and method section into paragraphs according to the “study population,” “the process of collecting information,” “covariate”, and “statistical analysis”.

(2) Page 2 line 83: how to group about maternal education level? please correct according to the first result: Crowding Index, Education, Return to Work

Response: We grouped maternal education level as University complete versus No University.  The mention of this grouping is now included in this sentence. Also ‘Return to Work’ is included in the same sentence because we also adjusted our analysis by ‘Return to Work’. This change is highlighted in yellow. In multivariate logistic regression analysis, only ‘Return to Work’ was adjusted for in the analysis while ‘Crowding Index’ and ‘Education’ weren’t adjusted for as only the former was found to be an operational confounder in the relationships examined. In sensitivity analyses we adjusted for all four covariates (maternal return to work, maternal education, crowding index and maternal age) in our logistic models and found that the results for the effects of overweight/obesity among 10-year-olds didn’t change from the ones which we reported in the paper that adjusted only for maternal return to work. These results are not shown in the paper but are available from the authors if required.

  • Advise to supplement Pvalue in table 1 and table 2.

Response: The p value is now added in table 1 and table 2.

(2) Insufficient adjustment of covariates in table 2, such as children’s age at breastfeeding cessation, maternal age, body mass index (BMI) during pregnancy, and social advantage index et al. This must be based on science and literature and not by choice.

Response: All the covariates suggested were not required to be adjusted while for some there were no data. For example, there was insufficient data to calculate social advantage index. So, maternal return to work, maternal education and crowding index were considered as proxies for social advantage. Only maternal return to work was found to be an operational confounder which performs stronger risk adjustment than classical confounding and so was included as a covariate in the model reported in table 2. This is already discussed in Methods. There was insufficient data to calculate children’s age at breastfeeding cessation and body mass index during pregnancy. So we couldn’t assess whether these variables were operational confounders. Maternal age wasn’t found to be an operational confounder as well and so wasn’t included in the model. As sensitivity analyses we adjusted for all four covariates (maternal return to work, maternal education, crowding index and maternal age) in our logistic models and found that the results didn’t change from the ones which we reported in the paper that adjusted only for maternal return to work (results not shown in the paper). These information are now added in Methods. The additions in text are highlighted in yellow.

Discussion:

(1) The first paragraph of discussion needs to concise chief conclusion and innovation of your article, not to list all the results again. And it is recommended to add the implications of main find.

Response: Thank you for recommending these changes. I have added the implications of main findings. The discussion of results and their comparisons with other studies are shortened. 

Reviewer 2 Report

1. The results of this paper did not support the title, the results should focus on the relationship between infant-feeding and childhood overweight/obesity;

2. The representation of the sample was not good, with 53.5% children overweight/obesity. So the design of this study should revised to a case-control study using starting formula ≤6 months or >6 months as group variable.

3. The method to calculated the sample size was not clear.

4.The discussion should be rewritten.

Author Response

"1. The results of this paper did not support the title, the results should focus on the relationship between infant-feeding and childhood overweight/obesity;

Response: The title we think is quite appropriate as we investigated both infant feeding patterns as well as the effects of early introduction to formula and solid foods on risk of overweight/obesity among ten-year-olds. The revised title based on suggestion of reviewer 1 is:

“Infant-feeding patterns and the effects of early introduction to formula and solid foods on childhood overweight or obesity among 10-years-olds in a low socioeconomic area of Lebanon: An exploratory analysis”

  1. The representation of the sample was not good, with 53.5% children overweight/obesity. So the design of this study should revised to a case-control study using starting formula ≤6 months or >6 months as group variable.

Response: We agree to this point. We have now added in Materials and Methods how we found that there were 53.5% ten-year-olds who were overweight or obesity as:

“We calculated the ten-year-old’s BMI from his/her weight and height reported by the mother. If a child’s BMI was at least equal to the 85th percentile of the distribution of BMI then he/she was classified as overweight or obese”.

The sample for our study was based on a cross-sectional survey conducted between July 2019 and September 2019 in Burj Hammoud comprising of mothers who were of Lebanese descent and during the time of interview had a 10-year old child who at birth was singleton, full term, normal weight and healthy. These are clearly mentioned in the first paragraph of the Materials and Methods section:

“We conducted a cross-sectional study between July 2019 and September 2019 in Burj Hammoud which is considered one of the poorest and densely populated districts in the Middle East [10]. It is located in Mount Lebanon which is adjacent to Beirut. We used convenience sampling to select the mothers for this study………...The selection criteria for our survey were that mothers must be of Lebanese descent and currently have a 10-year old child who at birth was singleton, full term, normal weight and healthy…………….”.

A cross-sectional study design which we adopted assumed that the exposure variables measured retrospectively did not change over time which is acceptable as these covariates contained information which were specific to a particular period of time (infancy) of the ten-year-olds lifetime and are comparable to historical covariates. All the regression analyses we performed pertain to a cross-sectional study design. We agree that a case-control study could have been conducted. We decided during the study planning that there would have been no or little advantage of conducting a case-control study because the temporal relationship between the supposed cause and effect cannot be determined by a case-control study.   Also, conducting a cross-sectional study of eligible mothers of ten-year-olds was logistically more convenient than conducting a case-control study. Finally, given the short time to respond to reviewer’s comments it would not be possible to implement the case-control design as it will involve changing most of the statistical analysis.

  1. The method to calculate the sample size was not clear.

Response: The second paragraph of Materials and Methods has stated clearly calculation of sample size. According to this, the minimum sample size required ignoring any nonresponse was 100, and we recruited 101 eligible mothers thus fulfilling the minimum requirement. The upper limit for sample size was set at 120 considering potential nonresponse which however out to be pretty low when the data was actually collected. The text from the manuscript follows:

“We did not require a large sample size to investigate our primary analytical re-search question as sample size estimations during the planning stage of the study showed that at least 100 participants would be adequate to detect significant effects for both introduction to formula and solid foods within 6 months respectively, on odds of childhood overweight/obesity, after adjusting for several potential confounders and fixing power and type 1 error at 80% and 5% respectively. To account for likely non-response, we targeted to recruit 120 women and ended up having 101 who met the selection criteria and agreed to participate, thus fulfilling the minimum sample size requirement”.

  1. The discussion should be rewritten."

Response: Thank you for this comment. We have thoroughly revised the discussion section taking into account your suggestion.

Reviewer 3 Report

Reviewer reports:

I found this paper presents important information that adds to the infant-feeding patterns knowledge pool, and its effect on childhood weight status among 10-years-olds in low socioeconomic status region.
I have few comments to be considered on the revised version

1. Principal comments:

Title:

(1) Please explain the difference between “infant feeding patterns” and “early introduction to formula and solid foods”. The meaning of infant-feeding patterns is not defined on this revised version.

Materials and method:

(1) In the material and method section, it is suggested to divide into paragraphs according to the “study population,” “the process of collecting information,” “covariate,” “statistical analysis” and so on.

(2) Page 2 line 83: how to group about maternal education level? please correct according to the first result: Crowding Index, Education, Return to Work

Results:

(1) Advise to supplement P value in table 1 and table 2.

(2) Insufficient adjustment of covariates in table 2, such as children’s age at breastfeeding cessation, maternal age, body mass index (BMI) during pregnancy, and social advantage index et al. This must be based on science and literature and not by choice.

Discussion:

(1) The first paragraph of discussion needs to concise chief conclusion and innovation of your article, not to list all the results again. And it is recommended to add the implications of main find.

2. Minor comments:

Title:

(1) “Childhood overweight or obesity” should be identified “childhood overweight or obesity among 10-years-olds”, because the design is a cross-sectional study.
Abstract:

(1) Page 1 line 11-12: the first sentence of abstract is adversative relation?

(2) “Future research needs to focus on conducting a larger study 24 by incorporating other low SES regions to confirm these relationships” In this section, “SES” does not recommend abbreviations.

Introduction:

(1) Page 2 line 29-30: Please list the evidence about infant-feeding patterns and childhood weight status in order to prove your hypothesis after the first sentence, which makes rigorous.

(2) Page 2 line 39-40: “it has been found that infants having an early introduction to formula or solid foods have higher likelihood of childhood overweight or obesity” is incomplete and must be corrected.

(3) In the introduction section, advise to complement study gaps in order to explain the study aims.

Materials and method:

(1) Page 2 line 62-64: “In Lebanon infant formulas …… buy them from dispensaries.” The approach might be biased, because poor infant-feeding patterns might be caught easily.

(2) Page 2 line 77-81: “Weight and length of children were not measured by anthropometric methods, but by using reports given by mothers.” How does the reporters judge childhood overweight/obesity? How does the author control the recall bias about history of feeding? please elaborate.

Author Response

I found this paper presents important information that adds to the infant-feeding patterns knowledge pool, and its effect on childhood weight status among 10-years-olds in low socioeconomic status region.
I have few comments to be considered on the revised version

  1. Principal comments:

Title:

  • Please explain the difference between “infant feeding patterns” and “early introduction to formula and solid foods”.The meaning of infant-feeding patterns is not defined on this revised version.

Response: By “Infant feeding patterns” we meant the dietary patterns of infants which include breastfeeding, formula, complementary feeding like semi-solid and solid foods. The timing of introduction to these specific diets is usually not of concern when we discuss infant feeding patterns. In contrast, by “early introduction to formula and solid foods” we meant infants who were introduced to formula and solid foods before their WHO recommended age of six months. The reference for the WHO recommendation in regards to this was already provided in our paper.

Materials and method:

  • In the material and method section,it is suggested to divide into paragraphs according to the “study population,” “the process of collecting information,” “covariate,” “statistical analysis” and so on.

Response: We have now divided the material and method section into paragraphs according to the “study population,” “the process of collecting information,” “covariate”, and “statistical analysis”.

(2) Page 2 line 83: how to group about maternal education level? please correct according to the first result: Crowding Index, Education, Return to Work

Response: We grouped maternal education level as University complete versus No University.  The mention of this grouping is now included in this sentence. Also ‘Return to Work’ is included in the same sentence because we also adjusted our analysis by ‘Return to Work’. This change is highlighted in yellow. In multivariate logistic regression analysis, only ‘Return to Work’ was adjusted for in the analysis while ‘Crowding Index’ and ‘Education’ weren’t adjusted for as only the former was found to be an operational confounder in the relationships examined. In sensitivity analyses we adjusted for all four covariates (maternal return to work, maternal education, crowding index and maternal age) in our logistic models and found that the results for the effects of overweight/obesity among 10-year-olds didn’t change from the ones which we reported in the paper that adjusted only for maternal return to work. These results are not shown in the paper but are available from the authors if required.

  • Advise to supplement Pvalue in table 1 and table 2.

Response: The p value is now added in table 1 and table 2.

(2) Insufficient adjustment of covariates in table 2, such as children’s age at breastfeeding cessation, maternal age, body mass index (BMI) during pregnancy, and social advantage index et al. This must be based on science and literature and not by choice.

Response: All the covariates suggested were not required to be adjusted while for some there were no data. For example, there was insufficient data to calculate social advantage index. So, maternal return to work, maternal education and crowding index were considered as proxies for social advantage. Only maternal return to work was found to be an operational confounder which performs stronger risk adjustment than classical confounding and so was included as a covariate in the model reported in table 2. This is already discussed in Methods. There was insufficient data to calculate children’s age at breastfeeding cessation and body mass index during pregnancy. So we couldn’t assess whether these variables were operational confounders. Maternal age wasn’t found to be an operational confounder as well and so wasn’t included in the model. As sensitivity analyses we adjusted for all four covariates (maternal return to work, maternal education, crowding index and maternal age) in our logistic models and found that the results didn’t change from the ones which we reported in the paper that adjusted only for maternal return to work (results not shown in the paper). These information are now added in Methods. The additions in text are highlighted in yellow.

Discussion:

(1) The first paragraph of discussion needs to concise chief conclusion and innovation of your article, not to list all the results again. And it is recommended to add the implications of main find.

Response: Thank you for recommending these changes. I have added the implications of main findings. The discussion of results and their comparisons with other studies are shortened. 

  1. Minor comments:

Title:

(1)“Childhood overweight or obesity” should be identified “childhood overweight or obesity among 10-years-olds”, because the design is a cross-sectional study.

Response: Thank you for this suggestion. “Childhood overweight or obesity” is now altered to “childhood overweight or obesity among 10-years-olds” throughout the text as well as in the title.

(2) Page 1 line 11-12: the first sentence of abstract is adversative relation?

Response: The first sentence in abstract is broken down into two. No link between early introduction to formula or solids and childhood overweight or obesity is discussed in these two sentences. So there is no more any adversative relation discussed at the start of the abstract.

(2) “Future research needs to focus on conducting a larger study 24 by incorporating other low SES regions to confirm these relationships” In this section, “SES” does not recommend abbreviations.

Response: SES is now replaced by socioeconomic in the abstract. Following this change and the ones discussed above the total word count still remains at 200 which was the limit set.

Introduction:

(1)Page 2 line 29-30: Please list the evidence about infant-feeding patterns and childhood weight status in order to prove your hypothesis after the first sentence, which makes rigorous.

Response: The evidence in support of the link between early introduction to infant formula and solid foods and higher risk of childhood overweight or obesity was already stated under references 7-9. This is now moved up after the first sentence in page 2 starting from line 30. Accordingly, these references are numbered 2-4 and a few subsequent reference numbers have changed.

(2) Page 2 line 39-40: “it has been found that infants having an early introduction to formula or solid foods have higher likelihood of childhood overweight or obesity” is incomplete and must be corrected.

Response: This sentence is revised and moved up after line 30 on page 2 as stated above. It stands as:In many countries, mostly of high and middle income level, it has been found that infants introduced to formula or solid foods within six months have higher likelihood of childhood overweight or obesity [2-4]”.

(3) In the introduction section, advise to complement study gaps in order to explain the study aims.

Response: Lines 50-52 under Introduction includes the study gaps:

“To our knowledge, this is the first study in Lebanon to examine whether early introduction to formula and solid feedings respectively are risk factors of childhood overweight or obesity”.

Materials and method:

  • Page 2 line 62-64: “In Lebanon infant formulas …… buy them from dispensaries.” The approach might be biased, because poor infant-feeding patterns might be caught easily.

Response: We are sorry that there has been a misunderstanding. What we wanted to convey in this sentence is that in Lebanon infant formulas can be purchased only from dispensaries unlike many other countries where these are sold mostly in supermarkets. There is no bias because people regardless of their children’s poor infant-feeding patterns have to buy these products from dispensaries.

 (2) Page 2 line 77-81: “Weight and length of children were not measured by anthropometric methods, but by using reports given by mothers.” How does the reporters judge childhood overweight/obesity? How does the author control the recall bias about history of feeding? please elaborate.

Response: We calculated the ten-year-old’s BMI from his/her weight and height. If a child’s BMI was at least equal to the 85th percentile of the distribution of BMI then he/she was classified as overweight or obese.  

We have now included the above in Methods section lines 87-89. Reduction of recall bias about history of feeding was already discussed on pages currently numbered 268-273 under Discussion section. It is stated: “A previous study showed that maternal recall of breastfeeding history of their children can be very accurate even 20 years after their birth [27]. Since the maternal recall period for our study was only 10 years, any underestimation of our observed associations which are due to maternal recall may be small. To minimize this further we excluded eligible mothers who did not confirm their response in regards to their children’s breastfeeding history after being asked to do so”.

Round 2

Reviewer 2 Report

The method section should be introduced in detail.

Author Response

Please see in the attached file.
